# Can reanalysis products outperform mesoscale numerical weather prediction models in modeling the wind resource in simple terrain?

Vincent Pronk[1], Nicola Bodini[1], Mike Optis[1], Julie K. Lundquist[1,2,3], Patrick Moriarty[1],
Caroline Draxl[1,3], Avi Purkayastha[1], and Ethan Young[1]

[1]National Renewable Energy Laboratory, Golden, Colorado USA
[2]Department of Atmospheric and Oceanic Sciences, University of Colorado Boulder, Boulder, Colorado USA
[3]Renewable and Sustainable Energy Institute, Boulder, Colorado USA

**Correspondence:** Nicola Bodini (nicola.bodini@nrel.gov)

**Abstract.** Mesoscale numerical weather prediction (NWP) models are generally considered more accurate than reanalysis products in characterizing the wind resource at heights of interest for wind energy, given their finer spatial resolution and more comprehensive physics. However, advancements in the latest ERA-5 reanalysis product motivate an assessment on whether ERA-5 can model wind speeds as well as a state-of-the-art NWP model—the Weather Research and Forecasting (WRF)

model. We consider this research question for both simple terrain and offshore applications. Specifically, we compare wind profiles from ERA-5 and the preliminary WRF runs of the Wind Integration National Dataset (WIND) Toolkit Long-term Ensemble Dataset (WTK-LED) to those observed by lidars at site in Oklahoma, United States, and in a U.S. Atlantic offshore wind energy area. We find that ERA-5 shows a significant negative bias ($\sim -1\,\mathrm{m\,s^{-1}}$) at both locations, with a larger bias at the land-based site. WTK-LED-predicted wind speed profiles show a limited negative bias ($\sim -0.5\,\mathrm{m\,s^{-1}}$) offshore and a

slight positive bias ($\sim +0.5\,\mathrm{m\,s^{-1}}$) at the land-based site. On the other hand, we find that ERA-5 outperforms WTK-LED in terms of the centered root-mean-square error (cRMSE) and correlation coefficient, for both the land-based and offshore cases, in all atmospheric stability conditions. We find that WTK-LED's higher cRMSE is caused by its tendency to overpredict the amplitude of the wind speed diurnal cycle. At the land-based site, this is partially caused by wind plant wake effects not being accurately captured by WTK-LED.

*Copyright statement.* This work was authored in part by the National Renewable Energy Laboratory, operated by Alliance for Sustainable Energy, LLC, for the U.S. Department of Energy (DOE) under Contract No. DE-AC36-08GO28308. Funding provided by the U.S. Department of Energy Office of Energy Efficiency and Renewable Energy Wind Energy Technologies Office and by the National Offshore Wind Research and Development Consortium under Agreement No. CRD-19-16351. The views expressed in the article do not necessarily represent the views of the DOE or the U.S. Government. The U.S. Government retains and the publisher, by accepting the article for publication,

acknowledges that the U.S. Government retains a nonexclusive, paid-up, irrevocable, worldwide license to publish or reproduce the published form of this work, or allow others to do so, for U.S. Government purposes.

# 1  Introduction

Wind energy development requires an accurate characterization of the wind resource at the heights swept out by commercial wind turbine rotor blades (Brower, 2012). Directly measuring the wind speed aloft for the extensive periods of time required for an accurate wind resource assessment can be challenging from both a technical and financial point of view. For land-based sites, there are several major factors that can pose limitations to the installation of tall meteorological towers or remote sensing devices, including complex topography, road access, availability of electrical power, and excessive cost. When considering offshore regions, where an unprecedented growth in installed wind capacity is currently taking place worldwide (Musial et al., 2016; Bureau of Ocean Energy Management, 2018), the challenges connected to having direct observations of hub-height wind speed are even more severe. Floating lidars represent a state-of-the-art source of offshore wind speed observations aloft (Carbon Trust Offshore Wind Accelerator, 2018; OceanTech Services/DNV GL); however, their prohibitive cost still severely limits their availability worldwide. As a result of this constrained scenario, numerical weather prediction (NWP) models at the mesoscale and reanalysis products are frequently used (Draxl et al., 2015; Hahmann et al., 2020; Dörenkämper et al., 2020; Optis et al., 2020) to characterize the wind resource at the heights of interest for wind energy development, for both land-based and offshore locations.

Reanalysis products are convenient to use given their global coverage and publicly available data. In general, reanalysis products incorporate global measurements of atmospheric variables to produce a 3D-gridded, hindcast, best estimate of the state of the atmosphere. Reanalysis products typically provide multiple decades of data and are regularly updated (Schwartz et al., 1999; Compo et al., 2011; Gelaro et al., 2017; Bloomfield et al., 2018). While very convenient for wind resource studies, the coarse spatial ($\sim$1 degree) and temporal resolution (usually 6 hours) can produce inaccurate estimates of the wind resource. Specifically, previous validation studies at land-based sites have led to a wide variety of uncertainties associated with the product used, the location, the vertical level, and the vertical and horizontal spatial approximation technique used (Kubik et al., 2013; Rose and Apt, 2015; Ramon et al., 2019; Molina et al., 2021). Offshore, reanalysis products generally have better skills, and they have been used to create atlases of either wind resource or wind energy potential. Zheng et al. (2018) quantified the global offshore wind resource using the ERA-interim reanalysis product (Dee et al., 2011), while Soares et al. (2020) recently evaluated the global offshore wind energy potential using the more recent ERA-5 product (Hersbach et al., 2020). However, validating such reanalysis predictions against hub-height observations has rarely been done because of the scarcity of offshore wind speed observations at the heights of interest for commercial wind development. Sheridan et al. (2020) recently validated three reanalysis products using data at one single height from a floating lidar in the U.S. Eastern Seaboard, and found that ERA-5 had the best performance out of the four considered reanalysis products. Similar validations have been performed in Northern Europe, especially focusing on low-level jet events (Kalverla et al., 2019; Hallgren et al., 2020).

By comparison, NWP models generally provide more accurate estimates of the wind resource but are much more expensive to run. NWP models use a large-scale atmospheric product, such as a reanalysis product, as external forcing, while using higher spatial and temporal resolution to simulate more comprehensive physics. Several studies have applied NWP models to wind resource assessment (an extensive review can be found in Al-Yahyai et al. (2010)), at a variety of temporal and spatial scales.

Draxl et al. (2015) developed an NWP model-based wind speed atlas for the continental United States, and similar efforts have been completed for the European continent (Hahmann et al., 2020; Dörenkämper et al., 2020). Recently, NWP models have also been used to create offshore wind resource assessment datasets (Rybchuk et al., 2021). By providing high-resolution wind speed data, NWP models are beneficial for assessing the wind resource at specific sites of interest for wind energy development.

However, the development of NWP-based wind resource datasets is computationally expensive, especially when considering the fine horizontal, vertical, and temporal resolutions, as well as long-term periods of record.

Within this context, the latest reanalysis product, ERA-5 (Hersbach et al., 2020), comes with significant advancements compared to previous products, in terms of both spatial (∼32-km horizontally) and temporal (1 hour) resolutions. These improvements motivate an analysis on whether ERA-5 is capable of modeling wind speeds with an accuracy comparable if not superior

to the state-of-the-art mesoscale NWP model—the Weather Research and Forecasting (WRF) model (Skamarock et al., 2019), as part of the initial runs for the WIND Toolkit Long-term Ensemble Dataset (WTK-LED). A similar analysis was performed for the WRF-based New European Wind Atlas by Dörenkämper et al. (2020), who found a significant negative bias for ERA-5. For our evaluation, we consider vertical profiles of wind speed up to 200 m and focus on two sites of primary importance for present and future wind energy development in the United States. As an example of land-based site, we consider the U.S.

Department of Energy's Atmospheric Radiation Measurement (ARM) Southern Great Plains (SGP) long-term measurement site near Lamont, Oklahoma. Several wind plants have been built near this location in the past decade, and an international field experiment (the American WAKE experimeNt, AWAKEN) is currently being planned at this site to better characterize the atmosphere-wake interactions (Moriarty et al., 2020). Offshore, we focus on two floating lidars recently developed by the New York State Energy Research and Development Authority (NYSERDA) along the U.S. Eastern Seaboard, where an unprece-

dented development of offshore wind energy in the United States is currently being planned (of Ocean Energy Management, 2021). We describe the datasets in Section 2, where we also introduce the performance metrics we adopt in the analysis. The results of the evaluation of both WTK-LED and ERA-5 are presented in Section 3, where we also focus on the variability of the assessed performance with height, wind direction, and atmospheric stability. Finally, we conclude our analysis and suggest future work in Section 4.

## 2 Data and Methods

While complex terrain likely remains too challenging for an accurate wind speed representation by ERA-5 and will be the subject of future work, here we focus our analysis on simple terrain. More in detail, we perform a reanalysis and mesoscale model validation at two locations—one on land and one offshore. At both sites, publicly available hub-height wind speed observations are used.

For our land-based test case, we focus on the ARM SGP Central Facility (C1) site near Lamont, Oklahoma (Figure 1). The SGP Central Facility site is located in a fairly flat area with an elevation ranging from just ∼270 meters (m)–390-m above sea level in the area surrounding the site. As a result, the land is used primarily for agricultural purposes. Several wind power plants were built in the area in the last decade, as shown in the map in Figure 1. For our analysis, we consider data from January 01,

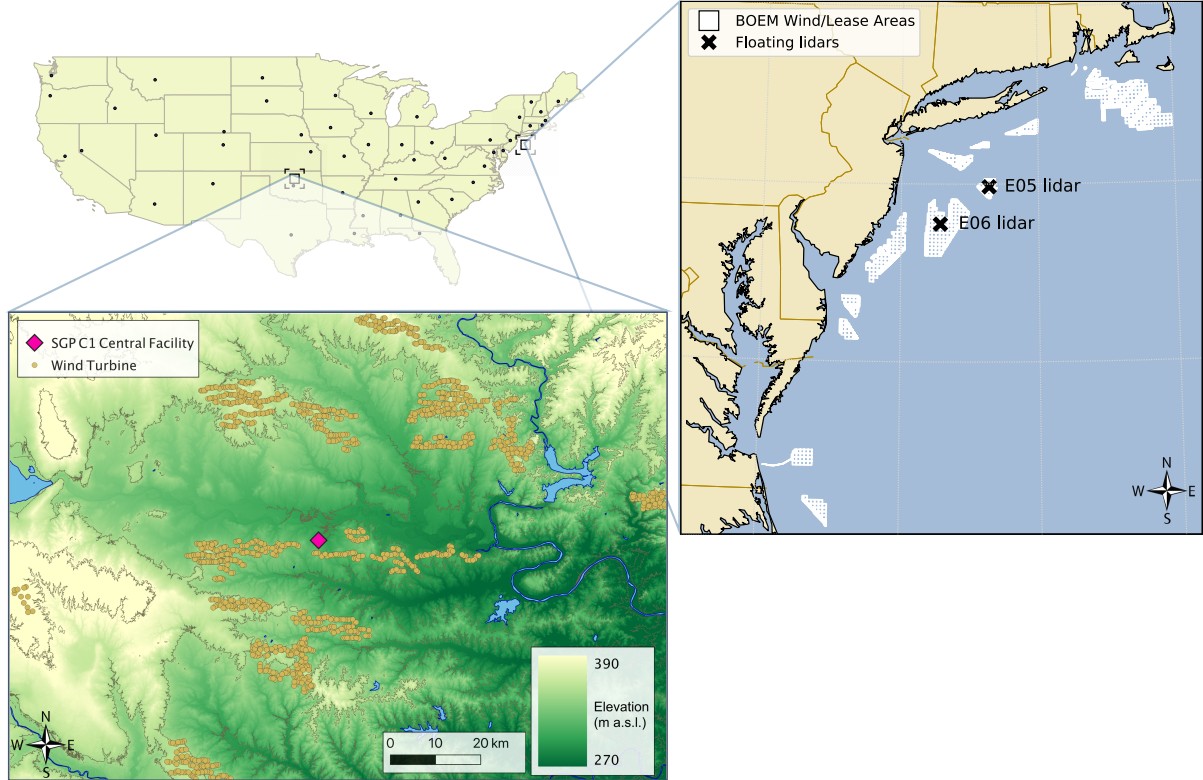

**Figure 1.** Map of the two sites considered in our validation analysis. Digital Elevation Model data courtesy of the U.S. Geological Survey.

2018 to December 31, 2018. While both ERA-5 and lidar observations at the site cover a much longer time period, preliminary
WTK-LED data for the region are available only for this one-year period.

Offshore, we use wind speed observations from two floating lidars along the U.S. Eastern Seaboard, where several wind energy lease areas have been planned (of Ocean Energy Management, 2021) for future offshore wind energy development (Figure 1). We consider data from September 01, 2019, to August 31, 2020 (lidar data are not available before this period and WTK-LED was not run after it).

## 2.1  Observations

At the SGP Central Facility site, we consider observations from a Halo Streamline lidar (Newsom, 2012). We obtain horizontal wind speed data from full 360° conical scans by the lidar, which were performed approximately every 15 minutes, with 1 minute needed to complete each scan. To obtain the horizontal wind speed from the line-of-sight velocity from these scans, we use the velocity-azimuth-display approach from Frehlich et al. (2006). In doing this, the horizontal wind field is assumed
to be homogeneous over the scan volume (Browning and Wexler, 1968). As in Bodini and Optis (2020), we discard any measurements that have a signal-to-noise ratio lower than −21 dB or higher than +5 dB, and any measurements from time periods

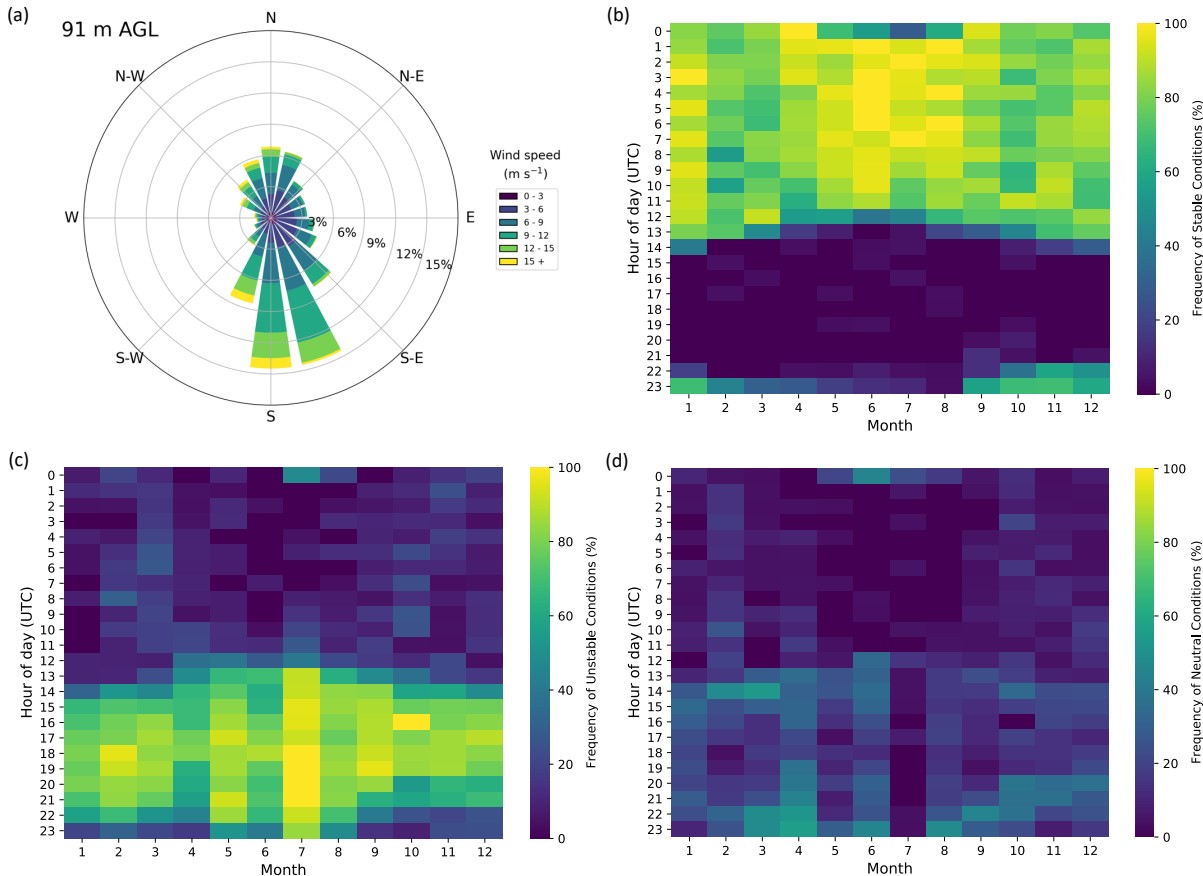

**Figure 2.** (a) Wind rose showing the distribution of wind speeds at 91-m AGL for 2018, using lidar observations at the SGP Central Facility site. (b) 24×12 heat map of the frequency of stable conditions at the site, classified in terms of the surface Obukhov length. (c) Same, but for unstable and (d) neutral conditions.

with precipitation that may significantly lower the accuracy of the measurements. The wind speed data are then averaged to obtain hourly average data. For our study, we consider data from 6 range gates, which correspond to heights of 65-, 91-, 117-, 143-, 169-, and 195-m above sea level. Data at lower heights are not used because of poor data quality. The analysis of the lidar
data reveals how the site experiences winds mainly from the south (see 91-m wind rose in Figure 2a), with more variability observed in winter months. Many wind plants were built around the SGP site in the last decade. At the Central Facility site, the closest wind turbines are about 3.5 km away, and Bodini et al. (2021) showed how the site is impacted by wind plant wakes for southerly flow, especially in nighttime stable conditions. We classify atmospheric stability using near-surface (4-m above ground level (AGL)) observations from a sonic anemometer at the C1 site. We calculate the Obukhov length as:

$$L = -\frac{\overline{T_v} \cdot u_*^3}{k \cdot g \cdot \overline{w'T_v'}} \tag{1}$$

where $k = 0.4$ is the von Kármán constant; $g = 9.81$ m s$^{-2}$ is the acceleration due to gravity; $T_v$ is the virtual temperature (K); $u_* = (\overline{u'w'}^2 + \overline{v'w'}^2)^{1/4}$ is the friction velocity (m s$^{-1}$); and $\overline{w'T_v'}$ is the kinematic virtual temperature flux (K m s$^{-1}$). All Reynolds decompositions are calculated with a 30-minute averaging period (De Franceschi and Zardi, 2003; Babić et al., 2012). We consider stable conditions for $0$ m $\leq L \leq 200$ m, unstable conditions for $-200$ m $\leq L < 0$ m, near-neutral otherwise. Figures 2b, c, and d show 24×12 heat maps of the frequency of stable, unstable, and neutral conditions, respectively. A clear diurnal pattern emerges, with stable conditions being typical of the nighttime hours, and unstable conditions occurring in daytime convective periods. Also, summer shows more extended unstable cases compared to winter months. On the other hand, near-neutral conditions are relatively rare at the site, occurring most often during morning and evening transitions. Finally, we consider measurements of sensible and latent heat fluxes from a flux station at the C1 site, which includes a sonic anemometer and an infrared gas analyzer, at 4 m AGL. Details on the calculation of the fluxes from these instruments are described in Fischer (2004), and follow the correction proposed in Webb et al. (1980).

Offshore, we consider observations of wind speed from two floating lidars off the coast of New Jersey. The New York State Energy Research and Development Authority (NYSERDA) recently deployed ZephIR ZX300M lidars and made their data publicly available (OceanTech Services/DNV GL). The lidar on buoy E05 is located at 39.97°N, 72.72°W; the buoy E06 lidar is located at 39.55°N, 73.43°W. The lidar observations are provided as hourly averages, after proprietary quality checks are applied to the data. Wind speed and wind direction data for both lidars are available every 20 m from 60-m to 200-m above sea level. At these sites, wind mainly flows from the southwest and northwest, as is the case further east in this region (Bodini et al., 2019, 2020) and it is generally stronger than what is observed at SGP, as shown in the wind rose at 100-m above sea level from lidar E05 in Figure 3a. Due to the lack of observations from which atmospheric stability metrics can be calculated, we use WTK-LED data to classify atmospheric stability as a function of the bulk Richardson number from 0 m–200 m above the surface. The bulk Richardson number is calculated as:

$$Ri_b = \frac{g}{\overline{\theta_v}} \frac{\Delta z \Delta \theta_v}{(\Delta U)^2 + (\Delta V)^2} \tag{2}$$

where $g$ is the gravitational acceleration, $\overline{\theta_v}$ is the average absolute virtual potential temperature across the considered layer of thickness $\Delta z$, $\Delta \theta_v$ is the virtual potential temperature difference across the layer, and $\Delta U$ and $\Delta V$ are the changes in the horizontal wind components across that same layer. We use values of $Ri_b > 0.025$ to classify stable conditions, $Ri_b < -0.025$ for unstable conditions, and all other values as near-neutral conditions. Figures 3b and c show the 24×12 heat maps of the frequency of stable and unstable conditions, respectively. While a clear diurnal pattern emerges when looking at similar plots at SGP, here we find little diurnal variability, but a strong seasonal cycle. Summer months show the most instances of stable conditions, while winter months show primarily unstable conditions. Finally, near-neutral conditions account for up to half of the cases in certain times and show little variability across both the diurnal and annual scales.

## 2.2 NWP model setup

At SGP, we use WRF model data for 2018 from the preliminary National Renewable Energy Laboratory's (NREL's) Wind Integration National Dataset (WIND) Toolkit Long-term Ensemble Dataset (WTK-LED), which will update the original WIND

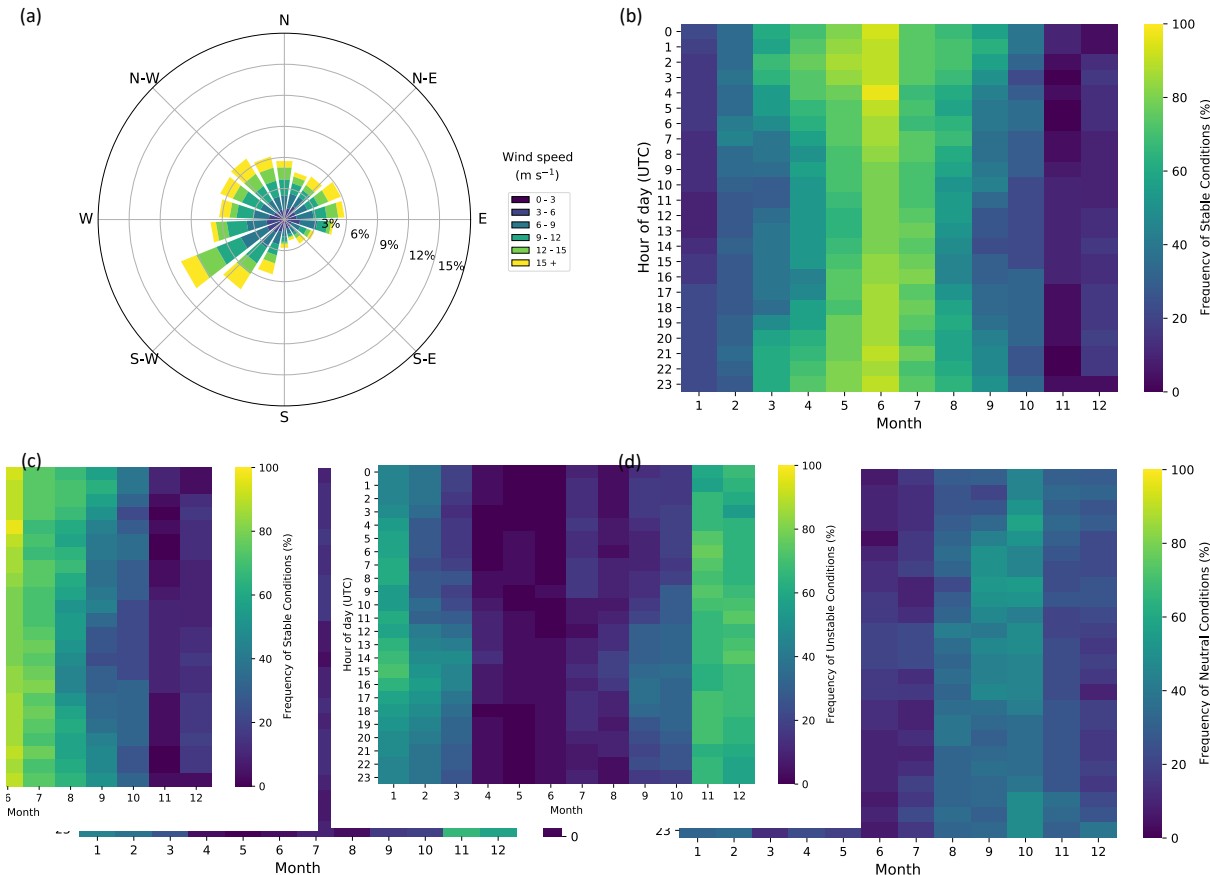

**Figure 3.** (a) Wind rose showing the distribution of wind speeds at 100-m AGL for September 2019 to August 2020, using observations from the E05 floating lidar. (b) 24×12 heat map of the frequency of stable conditions at the E05 lidar location, classified in terms of the WTK-LED-based bulk Richardson number calculated between 0 m and 200 m ASL. (c) Same, but for unstable and (d) neutral conditions. Results from the E06 lidar are included in the Supplementary Materials.

Toolkit (WTK) [Lieberman-Cribbin et al. (2014); King et al. (2014); Draxl et al. (2015)]. The main WRF attributes in the WTK-
145 LED setup are summarized in Table 1. All the main setups that have been shown to have a major impact on modeled wind speed
(e.g., the choice of the planetary boundary layer scheme and of the atmospheric forcing) are the same between the offshore
and land-based domains. For some other setups, different choices are made between the two domains in order to optimize and
tailor the numerical simulations to the specific needs of each domain. At both sites, the simulations are initialized every month,
and each simulation is initialized 2 days prior to and run up to 1 day after the end of each month. The first day of each monthly
150 run is used as spin-up time for the model, while the second and last days are used to combine monthly runs. Model output is
available at 5-minute resolution, and we average the data at hourly resolution to perform the validation analysis. We consider
data from the closest 2-km grid cell to the location of the lidar (the difference in terrain height between this WRF grid cell

and the actual lidar location is <3 m). To match the heights at which lidar observations are available, we linearly interpolate the WRF data from the two closest levels. Given the high near-surface resolution used (see Table 1), we expect this linear interpolation to introduce only a small additional error to the analysis.

**Table 1.** Key attributes of the WRF simulations in WTK-LED setup used in this study.

| Feature | Specification | |
|---|---|---|
| | Offshore | Land-based |
| WRF version | 4.2.1 | |
| Grid spacing | 6 km, 2 km (nested) | 2 km |
| Output time resolution | 5 minutes | |
| Vertical levels | 61 | |
| Near-surface-level heights (m) | 12, 34, 52, 69, 86, 107, 134, 165, 200 | |
| Atmospheric forcing | ERA-5 reanalysis | |
| Atmospheric nudging | Spectral nudging (6-km domain) applied every 6 hours | Spectral nudging applied every 3 hours |
| Planetary boundary layer scheme | Mellor-Yamada-Nakanishi-Niino Level 2.5 | |
| Microphysics | Ferrier | Morrison double-moment |
| Longwave radiation | Rapid radiative transfer model | |
| Shortwave radiation | Rapid radiative transfer model | |
| Topographic database | Global multiresolution terrain elevation data from the U.S. Geological Survey and National Geospatial-Intelligence Agency | |
| Land-use data | Moderate Resolution Imaging Spectroradiometer 30 s | |
| Cumulus parameterization | Kain-Fritsch (6-km domain) | None |
| Sea surface temperature product | Operational Sea Surface Temperature and Sea Ice Analysis (OSTIA) | None |

Offshore, we use WRF data from the offshore version of NREL's WTK-LED. A summary of the model setup is provided in Table 1. Similar to the land-based case, we select WTK-LED data from the closest grid cell to the location of each of the two floating lidars and linearly interpolate the WRF vertical levels to match the heights of the lidar data.

## 2.3 ERA-5 reanalysis

We use the state-of-the-art ERA-5 reanalysis product (Hersbach et al., 2020), to compare its skill in assessing wind resources with that of the WTK-LED product. ERA-5 provides hourly average data at 137 vertical levels and an ~31-km horizontal resolution. In our analysis, we consider vertical levels corresponding to heights of 54-, 79-, 106-, 137-, 170-, and 205-m above sea level. As done for WTK-LED, these heights are then linearly interpolated to match those of the lidar observations. We use data from the ERA-5 grid point which is closest in space to the considered lidars. Sheridan et al. (2020) recently showed that

selecting the closest grid point generally leads to better reanalysis performance compared to a linear interpolation of the four surrounding grid points, while Livingston and Lundquist (2020) used a bilinear interpolation. The coordinates of the selected ERA-5 grid point at SGP are 36.5°N and 97.5°W; offshore, we use data from the 40°N, 72.75°W grid point to compare with the E05 lidar data, and at 39.5°N, 73.5°W to compare with the E06 lidar data.

## 2.4 Performance metrics

To quantify the skills of WTK-LED and ERA-5 in predicting the observed wind resource, we calculate, at all of the considered heights, five performance metrics.

In general, it is important to decompose a model error into bias, which quantifies the difference between modeled and observed data; and random error, which quantifies the variability of the modeled data around the mean. We decompose the root-mean-square error (RMSE) into a bias component and an "unbiased" or "centered" component of RMSE (cRMSE), following the approach in Taylor (2001). We calculate bias as:

$$\text{Bias} = \bar{p} - \bar{o} \tag{3}$$

where $\bar{p}$ is the mean of the modeled (by either WTK-LED or ERA-5) estimates and $\bar{o}$ is the mean of the lidar observations. A perfect prediction would have a bias of 0. Next, we calculate the cRMSE as:

$$\text{cRMSE} = \left[ \frac{1}{N} \sum_{n=1}^{N} [(p_n - \bar{p}) - (o_n - \bar{o})]^2 \right]^{1/2} \tag{4}$$

where $N$ is the number of data points in the considered time series, and $p_n$ and $o_n$ indicate the time series values of modeled and observed wind speed, respectively. A perfect prediction would have a cRMSE of 0.

As a third performance metric, we calculate the square of the Pearson's correlation coefficient between observed and modeled wind. The correlation coefficient $r$ measures how strong the correspondence between two variables is, and it is calculated as:

$$r = \frac{\frac{1}{N} \sum_{n=1}^{N} (p_n - \bar{p}) - (o_n - \bar{o})}{\sigma_p \sigma_o} \tag{5}$$

where $\sigma_p$ and $\sigma_o$ are the standard deviations of the modeled and observed data, respectively. A perfect prediction would have a correlation coefficient of 1.

Next, we use the Earth-mover's distance (EMD), also known as the Wasserstein distance (Vaseršteĭn, 1969; Hahmann et al., 2020), which measures the difference between two distributions. EMD is calculated as the area between two cumulative distribution functions (here, modeled and observed wind speed). This metric is an improvement on the bias metric and will catch cases where bias may be zero, despite having different modeled and observed wind speed distributions. The distribution of a perfect prediction would have an EMD of 0.

Last, we compare the standard deviation of the observed wind speed with what is predicted by WTK and ERA-5.

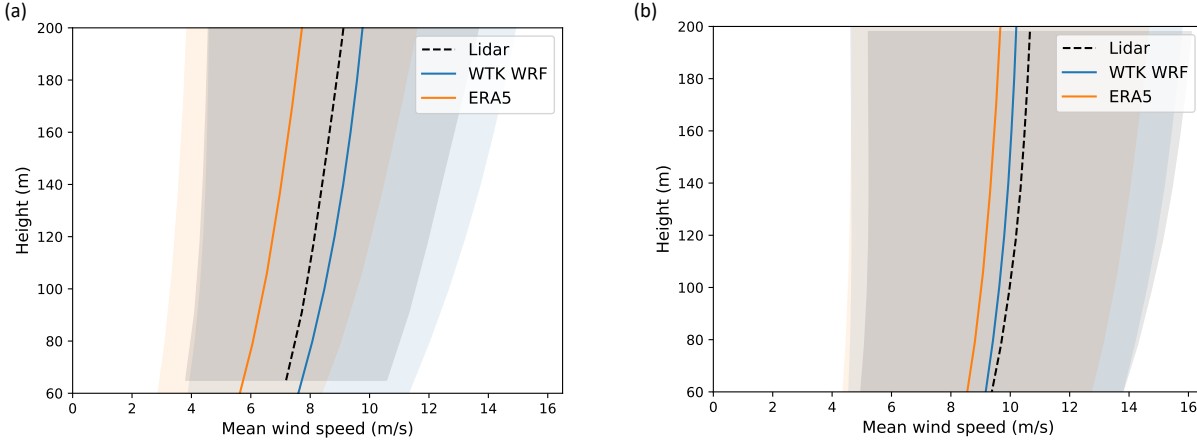

**Figure 4.** Mean vertical wind speed profiles for all three data sources at (a) the land-based SGP site and (b) the offshore E05 floating lidar. The shaded bands represent ± the standard deviation of the data.

## 3  Results

### 3.1  Mean performance

In Figure 4, we compare the mean wind profiles from all three data sources at SGP and the NYSERDA E05 lidar. (Results from the E06 lidar are included in the Supplementary Materials because no major differences between the results from the two lidars were found.) In each panel, the solid lines indicate the mean wind profile and the shaded bands around them represent ± the standard deviation of the data. On average, the wind resource is stronger offshore. At both sites, the ERA-5 mean wind profile underestimates the observed wind resource. Conversely, WTK-LED shows a limited overestimation of the mean wind

profile at the land-based site and a slight underestimation offshore. However, in all cases, a large variability emerges so that a more detailed investigation, beyond an annual average, is required.

We then consider the five mean performance metrics introduced in Section 2.4 for both WTK-LED and ERA-5 calculated at the two sites (Figure 5). The WTK-LED-predicted wind speed profiles show a limited positive bias ($\sim +0.5\,\mathrm{m\,s^{-1}}$) at the land-based location and a slight negative bias ($\sim -0.5\,\mathrm{m\,s^{-1}}$) offshore. However, ERA-5 shows a significant negative bias

at both locations, especially at SGP, where the bias is $\sim -1.5\,\mathrm{m\,s^{-1}}$. In general, we find little variability with height, with just a minor degradation of the bias with height for both WTK-LED and ERA-5. When considering the cRMSE, however, we find an opposite situation, with ERA-5 outperforming WTK-LED at both locations at all heights. We find satisfactory correlation at both sites, once again with ERA-5 providing marginally better values. The offshore location shows larger values ($r^2 > 0.85$ for both ERA-5 and WTK-LED at all heights), likely because of the positive effects of the simpler topography on

the skills of both data sources. Interestingly, we find a minor increase in $r^2$ with height, especially at SGP. When looking at the EMD, WTK-LED significantly outperforms ERA-5 at both sites, once again with the offshore site showing better results for both data sources compared to the land-based location. Finally, when comparing the modeled and observed wind speed

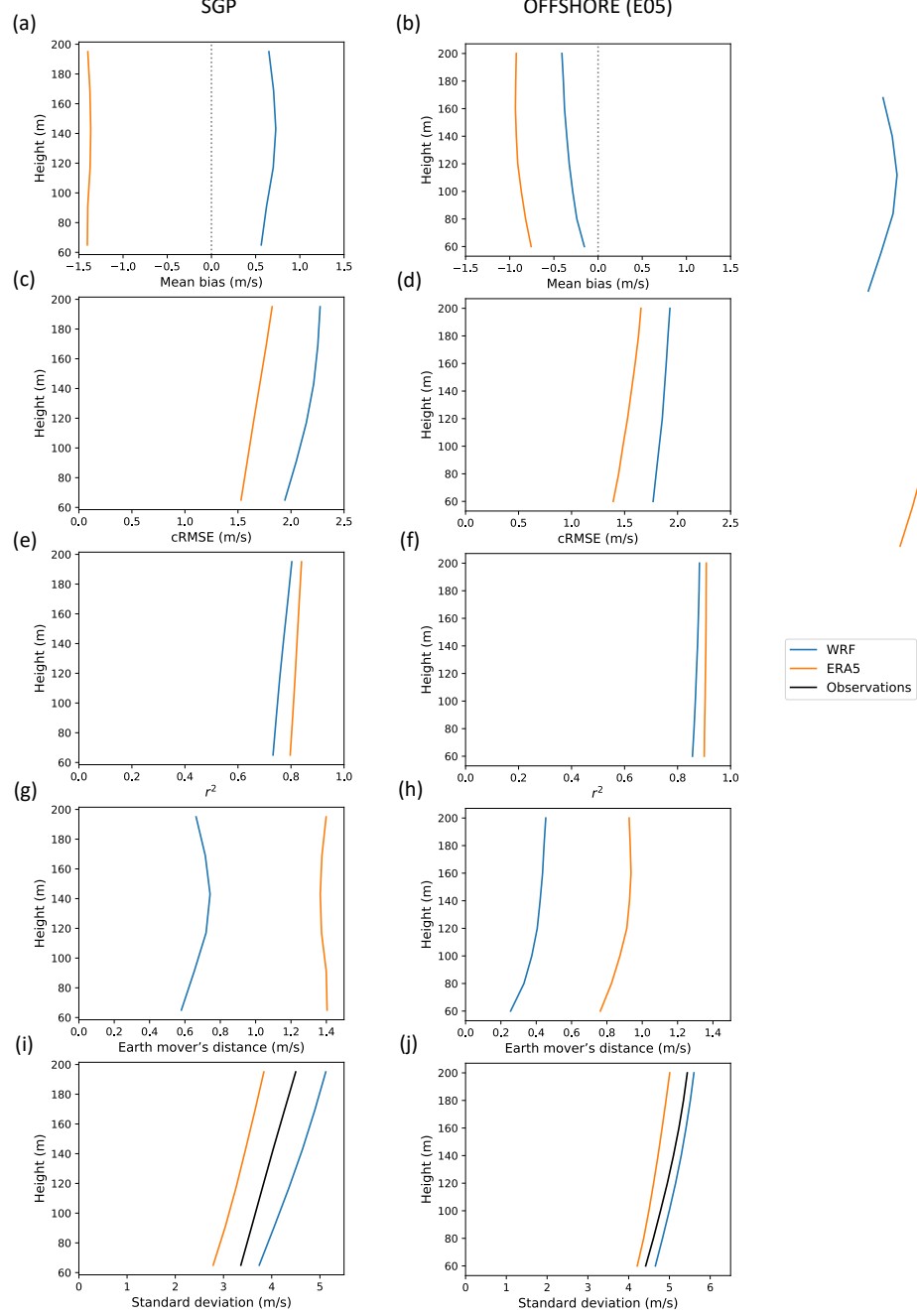

**Figure 5.** Vertical profiles of mean bias, cRMSE, $r^2$, EMD and wind speed standard deviation at the SGP C1 site (left) and the E05 lidar (right). Results from the E06 lidar are included in the Supplementary Materials.

standard deviations, we find no clear winner between WTK-LED and ERA-5 at SGP, whereas offshore WTK-LED provides
more accurate results. Given the difference in relative performance between WTK-LED and ERA-5 when considering different
metrics, we will investigate the impact of wind direction, atmospheric stability, diurnal, and seasonal cycles in the next sections
to investigate the potential reasons for such variability.

## 3.2  Impact of atmospheric stability

To assess whether the relative performance between ERA-5 and WTK-LED holds in all atmospheric stability conditions,
we segregate the data at both sites. As detailed in Sections 2.1.1 and 2.2.1, we classify atmospheric stability at SGP based
on the near-surface observed Obukhov length, while offshore we base our classification on the WTK-LED-modeled bulk
Richardson number, in absence of direct observations from which atmospheric stability parameters can be derived. We then
calculate the vertical profiles of the five performance metrics for each stability class at the two sites (Figure 6). The relative
performance between ERA-5 and WTK-LED observed from the mean profiles with no data segregation still holds in all stability
conditions: WTK-LED outperforms ERA-5 for bias EMD and, at the offshore site, standard deviation, while ERA-5 shows a
better performance when considering cRMSE and $r^2$. When analyzing how the performance of each data source varies with
atmospheric stability, interesting considerations emerge.

At SGP, WTK-LED shows the best agreement with observations in unstable conditions, with a near-zero bias and EMD at all
considered heights, and its lowest values of cRMSE. However, stable cases seem the most challenging to model for WTK-LED,
as also noticed by Smith et al. (2018, 2019) with respect to the challenges for WRF to accurately model the frequent nocturnal
low-level jets in the region. ERA-5 also performs well in unstable conditions, but ERA-5's stable cases outperform neutral
conditions, which show worse performance in terms of bias, cRMSE, and EMD. Little variability with stability emerges when
considering the $r^2$ metric.

At the offshore site, slightly different considerations apply. While unstable conditions still show the best performance in
terms of cRMSE, stable and neutral cases outperform unstable conditions when considering bias and EMD, with again little
variability in terms of correlation. For ERA-5, unstable cases show the best performance across all the considered metrics,
followed by neutral conditions and, last, stable periods.

## 3.3  Impact of wind direction

Different wind direction regimes could also have an impact on the relative performance of WTK-LED and ERA-5. In general,
we find that both WTK-LED and ERA-5 are capable of accurately representing the observed wind direction distributions at
the considered locations (see histograms in the supplementary materials). At SGP, the wind is mainly from the south or north
(wind rose in Figure 2), with a strong seasonal variability as summer months experience mostly southerly flow, whereas in
winter a broader range of wind directions is observed. At the offshore location, the winds are mostly from the northwest or the
southeast (wind rose in Figure 3), once again with a significant seasonal variability (as described in Bodini et al. (2019)), with
summer seeing mostly southwesterly winds, and winter experiencing mostly northwesterlies. Based on these dominant wind
regimes, we show in Figure 7 vertical profiles of the five performance metrics with data segregated by wind direction. At SGP,

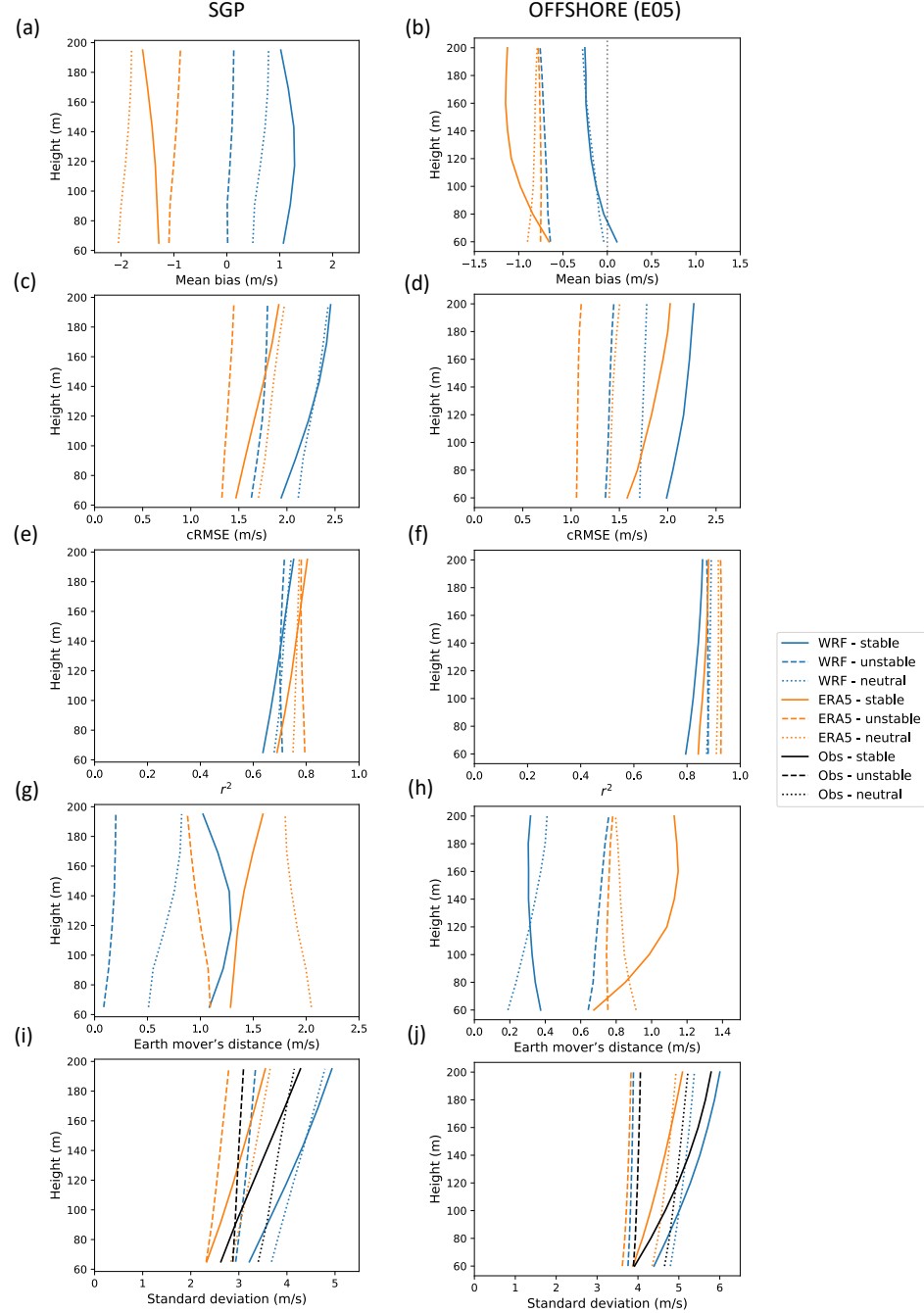

**Figure 6.** Vertical profiles of performance metrics segregated by atmospheric stability at the SGP C1 site (left) and at the location of the E05 lidar (right). Results from the E06 lidar are included in the Supplementary Materials.

we include northerly (315-45°) and southerly (135-225°) flow, at the E05 lidar we show results for northwesterly (270-360°) and southwesterly (180-270°) winds.

At SGP, both WTK-LED and ERA-5 show a better performance for northerly flow, when looking at bias, cRMSE and EMD. In terms of correlation, southerly flow cases are marginally better for both data sources, but with a very limited difference. As will be detailed in Section 3.5, the worse performance for southerly flow at the site can be connected to potential wind turbine and wind plant wake effects, which impact the SGP observational site for southerly winds.

At the offshore site, WTK-LED shows better skills when modeling southwesterly flow, whereas ERA-5 is more accurate at representing northwesterly flow. This is true across all performance metrics.

## 3.4 Impact of diurnal and seasonal variability

To further investigate the reasons for WTK-LED displaying a worse performance in its cRMSE and $r^2$ compared to ERA-5, we analyze the average diurnal cycle in wind speed at both locations (Figure 8). A clear diurnal variability emerges at both locations, with higher wind speeds occurring at night (SGP) or evening (offshore). At SGP, this variability is consistent with the frequent nocturnal low-level jets that have been observed at the site (Song et al., 2005; Greene et al., 2009). We find that ERA-5 well captures the amplitude of the observed diurnal cycle with a negative bias that remains nearly constant throughout the average day. By contrast, WTK-LED overestimates the amplitude of the average diurnal cycle, especially at the land-based location. At SGP, we find that WTK-LED significantly overestimates the nocturnal high wind speeds, whereas it slightly underestimates the daytime wind regime. Offshore, a nearly opposite situation occurs as WTK-LED exhibits skill in predicting the strong nocturnal winds, but marginally underestimates the weaker daytime winds. This exaggeration of the diurnal cycle by WTK-LED leads to its worst performance, compared to ERA-5, when considering the cRMSE and correlation coefficient at both locations.

To further break down the temporal variability of the relative performance of WTK-LED and ERA-5, we look at the diurnal and seasonal variability of bias, cRMSE, $r^2$, and EMD at both test sites. To do so, we build $24 \times 12$ heat maps of the four metrics by partitioning data by both hour of day and month. We show results for the land-based test case in Figure 9; the offshore test case is shown in Figure 10. We show results at 91-m AGL at SGP, at 100-m offshore, considered as a proxy for wind turbine hub-height, and we note that no significant variability of the metrics with height was found. At SGP, the analysis of bias confirms what was seen in Figure 8—that ERA-5 displays a negative bias for all months and hours, whereas WTK-LED shows mostly a positive bias during the night and a negative bias during the daytime, thus confirming that WTK-LED overestimates the diurnal cycle throughout the year. While the daytime bias performance of WTK-LED does not change significantly throughout the year, we find a larger positive bias in summer nights, whereas in winter the bias is smaller. As mentioned in the previous section, this could connect to the seasonal variability of the wind regimes at the site, where summer months show dominant winds from the south, whereas winter experiences a larger variability of wind directions. When looking at the differences in the cRMSE between WTK-LED and ERA-5, we find more instances where WTK-LED shows higher cRMSE values than ERA-5 during the nighttime than the daytime. As already noted for bias, we find a worse WTK-LED performance during nighttime in summer months also in terms of cRMSE, correlation, and EMD. In general, WTK-LED shows significantly better performance

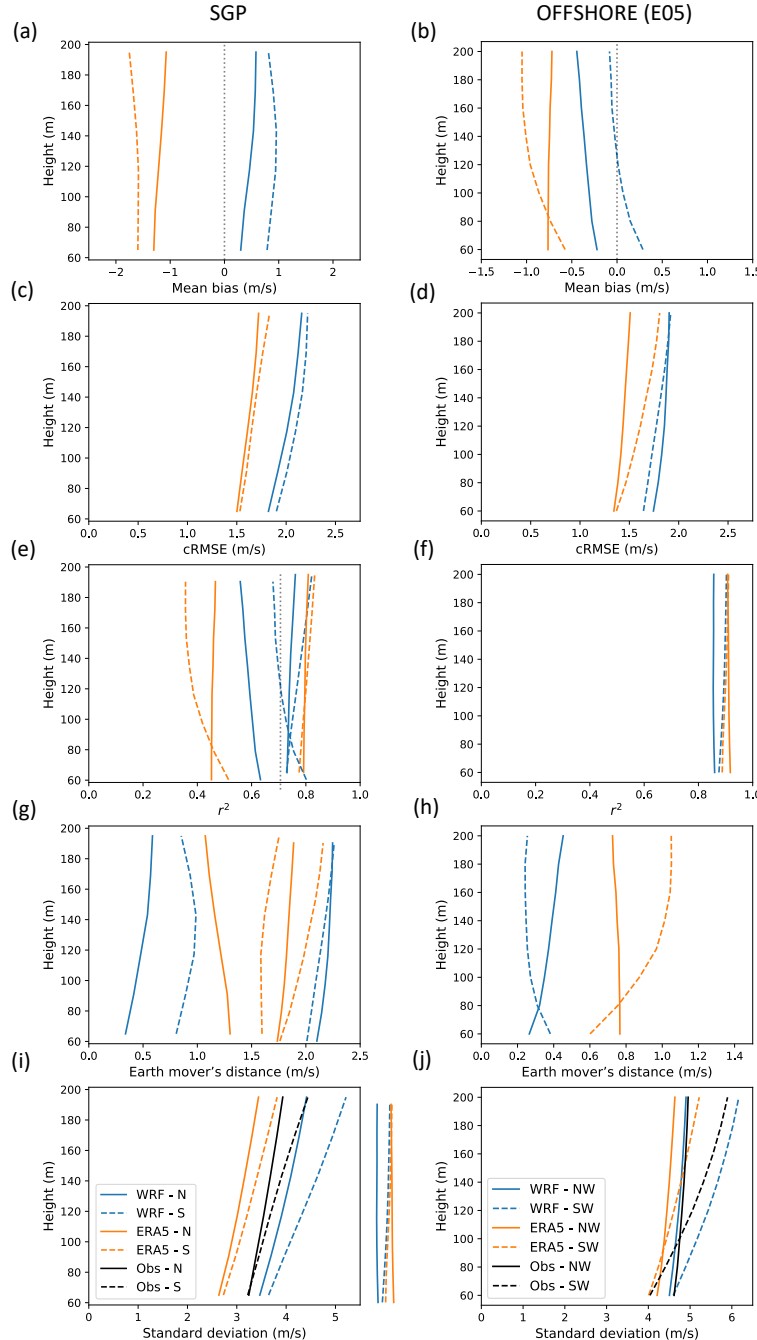

**Figure 7.** Vertical profiles of performance metrics segregated by dominant wind direction regimes at the SGP C1 site (left) and at the location of the E05 lidar (right). Results from the E06 lidar are included in the Supplementary Materials.

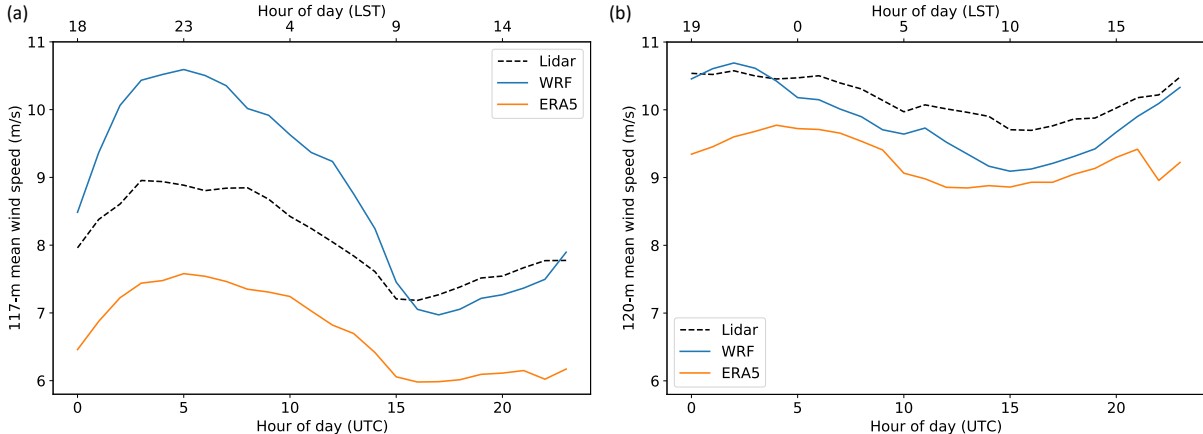

**Figure 8.** Average diurnal cycle of the ∼120-m wind speed from lidar, WRF, and ERA-5 at (a) the SGP C1 site and (b) the location of the E05 lidar. Results from the E06 lidar are included in the Supplementary Materials.

than ERA-5 in terms of EMD during the daytime, whereas there are several instances during the nighttime, especially in the late spring and summer months, where ERA-5 outperforms WTK-LED.

At the offshore lidar location, ERA-5 shows a negative bias for all months and hours, similar to what was seen at SGP. WTK-LED displays more occurrences of positive biases in spring and summer months, especially at night, whereas a limited negative bias is observed during the winter months at all hours. The overestimation of the observed diurnal cycle by WTK-LED is therefore more typical of summer months. However, little variability emerges when considering the relative performance of WTK-LED and ERA-5 in terms of the cRMSE, at both diurnal and annual scales. In fact, in the majority of cases, WTK-LED displays a higher cRMSE than ERA-5, but without any clear seasonal or diurnal pattern, thus making the interpretation of the results murkier compared to the land-based case. Looking at the heat map for the EMD, a consistent seasonal or diurnal pattern is not clear, either. However, WTK-LED generally outperforms ERA-5 in terms of EMD, with its best performance in the spring.

### 3.5 Impact of wind plant wakes at land-based site

As already mentioned, the land-based location, during 2018, is influenced by the presence of a large number of wind power plants in its vicinity (see map in Figure 1). As shown by Bodini et al. (2021), wakes from wind plants in the vicinity affect the lidar measurements at the C1 location when the wind is flowing from the south, and wake effects tend to be stronger in stable conditions. Because the WTK-LED predicts stronger winds at night than the lidar observes, some of the exaggerated diurnal cycle could be due to the fact that the WTK-LED does not incorporate effects from wind power plants. To further investigate this possibility, Figure 11 shows the hub-height wind speed average diurnal cycle at SGP for southerly flow (i.e., when the wind turbines are directly upwind of the lidar, panel (a)) and northerly flow (i.e., when the lidar measurements are unaffected by the wind power plants, panel (b)). We find that WTK-LED overestimates the hub-height wind speed especially for southerly flow,

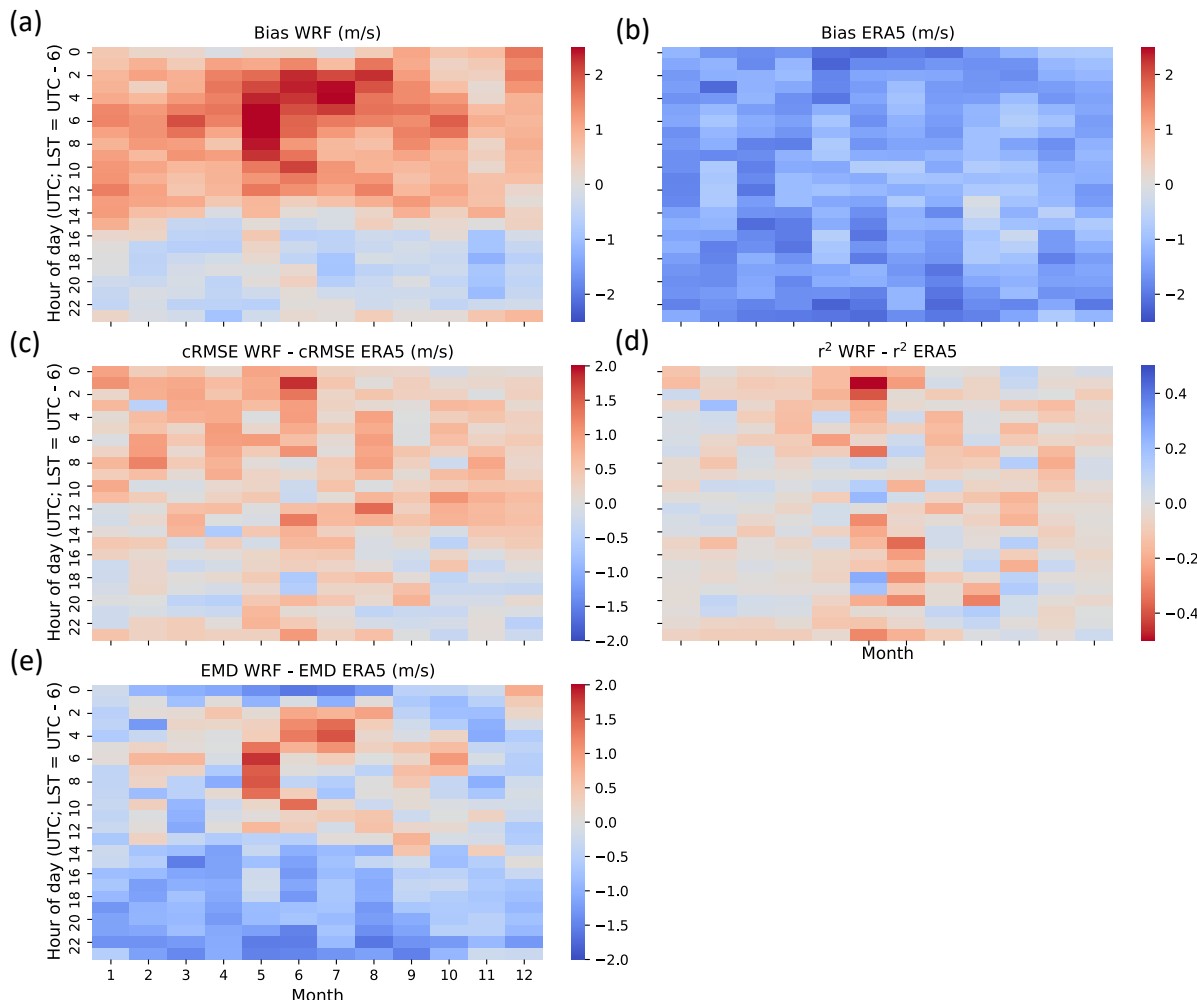

**Figure 9.** 24×12 heat maps at the SGP C1 site showing the diurnal and seasonal variability in bias, cRMSE, $r^2$, and EMD for the 91-m wind speed.

while only a marginal overestimation is observed for the less frequent northerly flow. This result suggests how wind power plant wakes, which are not represented by WTK-LED, could contribute to its strong overestimation of wind speed during stable conditions, which is consistent with the results presented in the previous sections. Based on the quantification of the wake impacts on the SGP C1 lidar in Bodini et al. (2021), wakes reduce wind speeds at C1 by up to 0.7 m s−1, and can therefore explain about a third of the WRF nighttime overestimation of wind speed at the site.

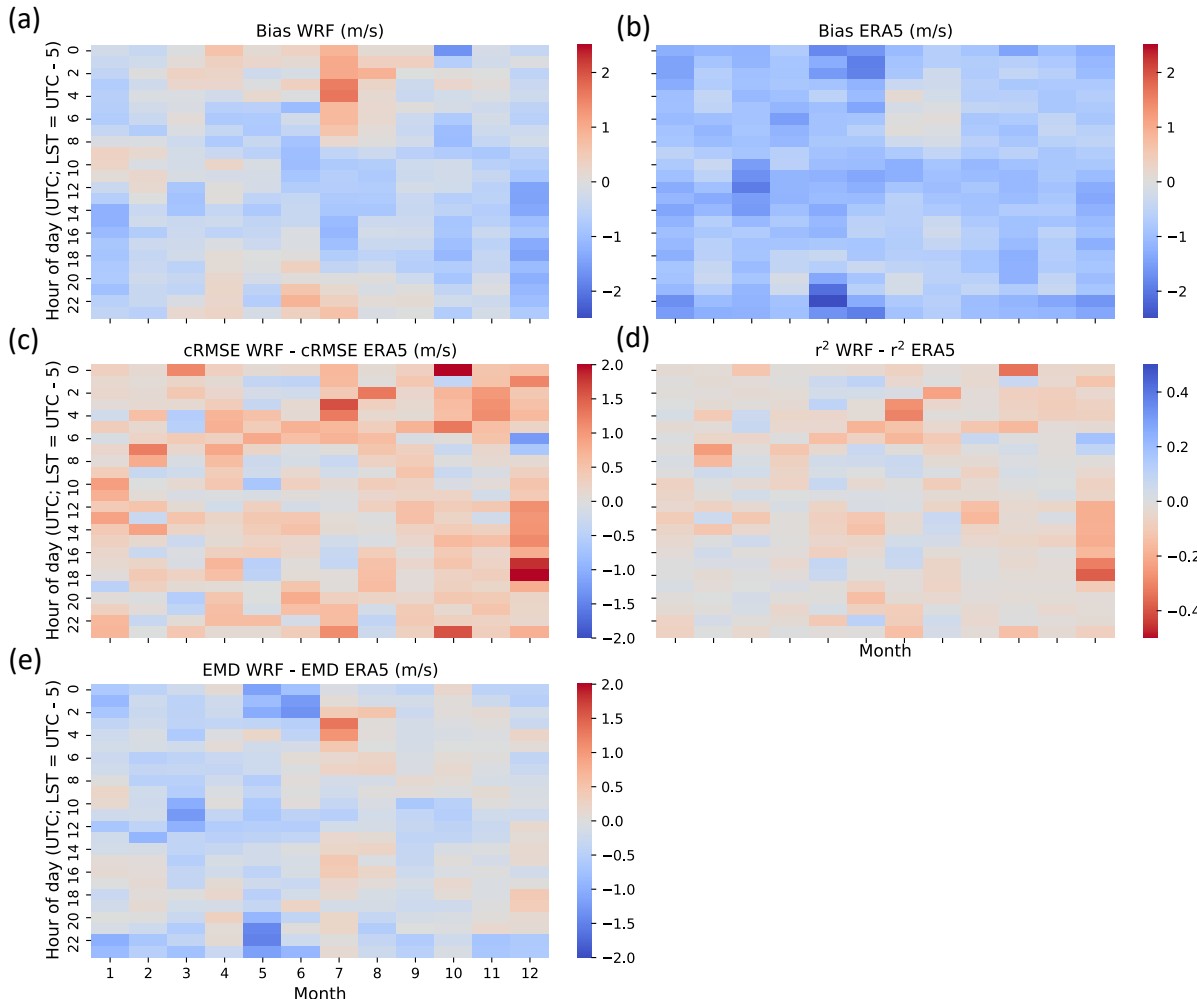

**Figure 10.** 24×12 heat maps at the location of the E05 lidar showing the diurnal and seasonal variability in bias, cRMSE, $r^2$, and EMD for the 100-m wind speed. Results from the E06 lidar are included in the Supplementary Materials.

## 3.6 Impact of surface fluxes at land-based site

The ability of WTK-LED of accurately representing soil moisture could also contribute explaining the overestimation of the diurnal cycle at SGP, as the variability of surface fluxes can change atmospheric turbulence and, as a consequence, drag (Geernaert, 1990). In Figure 12, we compare the average diurnal cycle in sensible and latent heat fluxes from near-surface observations at SGP and from the WTK-LED. We find that WTK-LED overestimates the sensible heat flux during the day, while at night the WTK-LED prediction is, on average, quite accurate. The diurnal cycle of latent heat flux, on the other hand, is quite accurately modeled by WTK-LED throughout the day, on average, with a minor delay compared to the observed one. These results suggest that the exaggerated strong sensible heat flux by WTK-LED during the day (potentially caused by issues

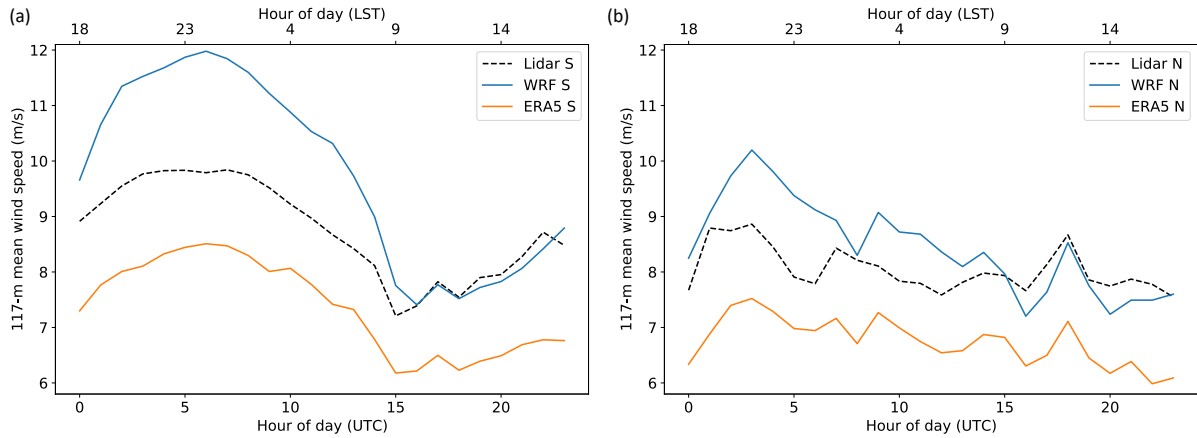

**Figure 11.** Average diurnal cycle of the 117-m wind speed from lidar, WRF, and ERA-5 at the SGP C1 site for (a) southerly (112°-196°) and (b) northerly (315°-45°) wind flow.

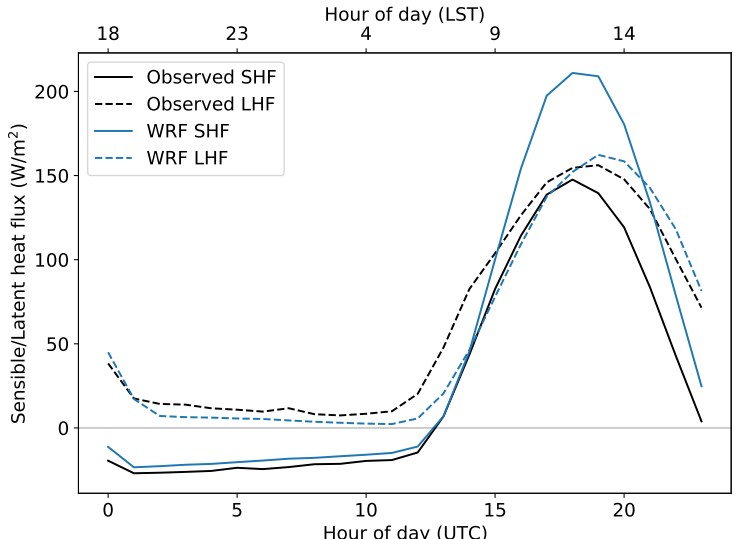

**Figure 12.** Observed and WTK-LED-modeled average diurnal cycle of sensible and latent heat fluxes at the SGP site.

in the WTK-LED land use/land cover) is consistent with the WTK-LED's slight underprediction in the mean diurnal cycle of hub-height wind speed between 16-23 UTC, as a strong sensible heat flux would increase turbulence and drag. Soil moisture seems instead accurately modeled in WTK-LED given the relatively accurate latent heat flux values, so that the effects on the exaggerated diurnal cycle in hub-height wind speed are likely negligible Xia et al. (2021).

315

## 4    Summary and Conclusions

Accurate characterization of the wind resource aloft is a necessity for wind energy development. At land-based locations, direct observations of the wind resource at hub height are often challenging to collect due to a variety of reasons, including cost, complex topography, road access, and availability of electrical power. Offshore, collecting direct measurements of wind speeds aloft is even more challenging. Thus, NWP models and reanalysis products are often used to characterize the wind resource in the locations of interest for wind energy development.

Using one year of lidar data at both land-based and offshore test sites, we evaluate the WRF model as run in the WTK-LED setup and the ERA-5 reanalysis product in their wind resource assessment skills. To evaluate each data product, we calculate five model performance metrics—bias, cRMSE, $r^2$, EMD, and a comparison between modeled and observed standard deviation. WTK-LED shows a smaller bias than ERA-5 at both the considered locations for all of the stability conditions. However, ERA-5 outperforms WTK-LED in terms of cRMSE for all stability cases both at the land-based and offshore sites. A potential explanation for this underperformance of WTK-LED in terms of cRMSE is WTK-LED's exaggeration of the average diurnal cycle at both sites, but especially at the land-based one. In fact, when considering the diurnal variability of the WTK-LED bias, we find that WTK-LED generally shows a positive bias at night and more instances of negative biases during the daytime. At the land-based site, WTK-LED mostly overestimates the nighttime wind speed for southerly flow, when the location is impacted by wind plant wakes, which are not captured in WTK-LED. Additionally, the WTK-LED negative bias during the day is instead connected to an overestimation of the sensible heat flux at the site. On the other hand, ERA-5 is capable of well capturing the amplitude of the daily cycle in hub-height wind speed at the considered locations, albeit with a relatively constant negative bias throughout the diurnal cycle. Both WTK-LED and ERA-5 show high correlation offshore, while at the land-based site the correlation is marginally reduced, likely because of the increased complexity in modeling the wind flow in conjunction with topographic effects. Analysis at both locations shows ERA-5 having a slightly stronger correlation than WTK-LED. Based on the analysis of the EMD, the wind speed distributions predicted by WTK-LED better match the observed distributions compared to the ERA-5 data in all stability conditions. Also, WTK-LED better matches the observed standard deviation of wind speed at the offshore site, whereas WTK-LED and ERA-5 have a comparable performance at SGP.

Our results show how there is not a clear and universal winner between WRF (in the WTK-LED setup) and ERA-5 when assessing their skills for wind resource assessment at these two locations, offshore and flat terrain on land. However, when weighting the relative performance of the two data sources, it is worth noting how bias correction techniques have been successfully applied in the wind energy sector (Stoffelen, 1998; Costoya et al., 2020). With this in mind, we can expect that the worse ERA-5 performance in terms of bias would be easier to accommodate when compared to the WTK-LED underperformance in terms of random error (cRMSE) and correlation, with the caveat that observations of the wind resource, which might be challenging and/or expensive to obtain, are needed for a successful bias correction. On the other hand, it is worth emphasizing that WTK-LED offers data at a finer spatial and temporal resolution, which represents an essential advantage over reanalysis products for specific wind energy related applications, such as grid integration analyses (Archer et al., 2017) and in locations with complex terrain. Clearly, it is important to stress that our results are specific to the sites considered in the analysis,

which are both characterized by simple topography. Future work can replicate our proposed validation in more complex terrain, where the coarser resolution of the reanalysis products is likely to have a severe negative impact on their skills in accurately representing the wind flow at hub height. Such analyses could provide additional understanding about why the WTK-LED WRF setup struggled, in our analysis, in well representing the wind speed diurnal cycle aloft. Finally, follow-on work will explore whether representing wind power plants in WRF improves the WRF performance in the vicinity of active wind power plants (e.g., by using the WRF Wind Farm Parameterization (Fitch et al., 2012; Tomaszewski and Lundquist, 2020)).

*Data availability.* Observations at the SGP site are publicly available at https://www.arm.gov/capabilities/instruments/dl. The NYSERDA lidar observations are publicly available at https://oswbuoysny.resourcepanorama.dnvgl.com. ERA-5 data are publicly available from the ECMWF's MARS archive. The WTK-LED data for the offshore domain are publicly available at https://maps.nrel.gov/wind-prospector/. The WTK-LED data for the land-based site will be available to the public in the future. The open-source WRF model was used for the numerical weather prediction simulations.

*Author contributions.* NB, MO, PM, and JKL envisioned the analysis. MO ran the offshore WRF simulations. CD, AP, and EY ran the land-based WRF simulations. VP analyzed the data, in close consultation with NB, and with general guidance from MO, PM, and JKL. VP and NB wrote the manuscript. All authors provided feedback on the paper draft.

*Competing interests.* The authors declare that they have no conflicts of interest.

*Acknowledgements.* This work was supported in part by the U.S. Department of Energy, Office of Science, Office of Workforce Development for Teachers and Scientists under the Science Undergraduate Laboratory Internship program. This research was performed using computational resources sponsored by the U.S. Department of Energy's Office of Energy Efficiency and Renewable Energy and located at the National Renewable Energy Laboratory.

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
