# Peer review of "Can reanalysis products outperform mesoscale numerical weather prediction models in modeling the wind resource in simple terrain?"

_Wind Energy Science, 2021_

## Author Comment (AC1)

*In this document, the reviewer's comments are in black, the authors' responses are in red.*

The authors thank the reviewer for their thoughtful and productive comments.

The paper presents a study of how well the ERA-5 reanalysis dataset and a WRF-based dataset can represent the wind conditions at two locations in North America for one year and an assessment of whether ERA-5 is sufficiently good for estimating wind resources in simple terrain, or whether mesoscale modeling is required.

The paper is well written and introduces the problem and the state-of-the-art well too, but misses references to a few recent studies that are relevant (see one specific example in general comments below). The figures in the paper and the accompanying descriptions of the results are easy to read and understand. The scope of the study is quite narrow, representing just two specific locations. However, both sites are of high relevance for wind energy and represent two distinct and relevant wind climates. Although no clear-cut answer is given to the question in the title of the paper, important results and their implications are discussed.

Four metrics based on wind speed are used to judge the model's performances: bias, centered root-mean-square error (cRMSE), Pearson correlation coefficient, and Earth Movers Distance (EMD). I believe the paper could benefit from including additional wind-energy-relevant metrics, for example, wind direction metrics, such as directional RMSE or directional EMD, and/or wind power metrics (power density or power production estimate).

All in all, I found the paper interesting and valuable and would recommend accepting it with minor revisions.

**Specific comments**

- P1L10 - I wonder if it actually is surprising that ERA-5 has a higher correlation, for hourly averages, than WTK-LED, perhaps choose a more neutral statement or expand on why it is surprising. From my experience, when looking at one point, reanalysis datasets and coarse mesoscale data often have a higher correlation than high-resolution mesoscale data due to higher variance (temporally and spatially) and phase errors
  We have replaced the word "surprisingly" with "on the other hand".

- P3L60-63 - I think you should reference previous comparisons between ERA-5 and WRF-based datasets for wind resource assessment accuracy, such as Dörenkämper et al. (2020), which you cited earlier in the introduction, which evaluated the models against a large number of masts in varying levels of terrain complexity in Europe and found significant underestimation of wind resources by ERA-5
  We have added the following sentence: "A similar analysis was performed for the WRF-based New European Wind Atlas by Dörenkämper et al. (2020), who found a significant negative bias for ERA-5."

- P8 table 1 - If possible, it would be good to provide the references to the different datasets and WRF physics options
  We have added the following comment: "All the main setups that have been shown to have a major impact on modeled wind speed (e.g., the choice of the planetary boundary layer scheme and of the atmospheric forcing) are the same between the offshore and land-based

domains. For some other setups, different choices were made between the two domains in order to optimize and tailor the numerical simulations to the specific needs of each domain.". Papers dedicated to the presentation of the new WRF datasets and their setups will follow as soon as all the planned regions are completed.

- P8L150-151 - "Confirmed" sounds as if it matches expectations or confirms previous studies showing that, is that the case? I am not convinced that nearest-neighbor interpolation has been shown to definitely be better in most cases
We have changed this to "showed".

- P14-15 Figure 8-9 - Why was the blue-to-red colormap flipped for the correlation coefficient subplot? I found it a bit confusing
The colormap was flipped compared to panels c and e because for cRMSE and EMD, a negative value for the difference between the metric from WRF and ERA-5 will indicate that WRF outperforms ERA-5, whereas for the correlation coefficient the opposite is true. In other words, we wanted to be consistent with the idea that red colors in all three panels will show that ERA-5 outperforms WRF, and vice versa for blue colors.

- P16L261 - Is wind power plant wakes represented at all by WTK-LED? if not I would change it accordingly, now it sounds as if partly represents wakes. In the last line, the conclusions (P17L295) leads me to believe that wakes are not resolved (yet) in WTK-LED
Thanks for letting us notice this was not clear. We have rephrased the sentence as "This result suggests how wind power plant wakes, which are not represented by WTK-LED, might contribute to its strong overestimation of wind speed during stable conditions."

- P17L274 - If possible please also offer an explanation, or perhaps just a speculation of the potential explanation, for the exaggerated average diurnal cycle. Perhaps it stems from the PBL and SL schemes used?
Unfortunately, we do not have a potential explanation at the moment. At the offshore site, we have seen the same behavior also when considering different WRF setups (in terms of different reanalysis product, PBL scheme, SST product, and SL scheme). We have currently in progress an analysis of a similar validation in complex terrain, which might help finding the reasons for the observed variability.

- P17L285 - Bias correction techniques are indeed valuable, but I think it is important to stress that they require observations or another reference dataset, known to do well at the site. Part of the motivation in your paper is that ERA-5 and WRF can serve as a cheap alternative to observations
We have added the following comment: "With this in mind, we can expect that the worse ERA-5 performance in terms of bias would be easier to accommodate when compared to the WTK-LED underperformance in terms of random error (cRMSE) and correlation, with the caveat that observations of the wind resource, which might be challenging and/or expensive to obtain, are needed for a successful bias correction."

- P17 data availability - Please state whether WTK-LED data can be obtained, and if so from where

We have added the following statement: "The WTK-LED data for the offshore domain are publicly available at https://maps.nrel.gov/wind-prospector/. The WTK-LED data for the land-based site will be available to the public in the future."

**Technical corrections**

- P4L92 - I would suggest using a consistent minus-sign throughout the paper, $-$21 dB instead of -21 dB, etc
  Changed.

- P5 Figure 2 and P5L108 - I would suggest 24 $\times$ 12, e.g. using latex \$\textbackslash times\$
  Changed.

- P5L105 - friction velocity and temperature flux units seem to have too much space between letters
  Changed.

- P9L173 - question mark in cite parenthesis, perhaps a reference was not compiled correctly?
  Fixed.

- P14-15 Figure 8-9 - Subplot letters missing
  Added.

---

## Author Comment (AC2)

*In this document, the reviewer's comments are in black, the authors' responses are in red.*

The authors thank the reviewer for their thoughtful and productive comments.

**Summary**

This study investigates the question whether a current state-of-the-art re-analysis product ERA-5 is sufficiently good to replace mesoscale models for wind resource assessments in simple terrain. Although the study doesn't provide a definite answer to the question, it provides a good contribution to the scientific community dealing with these type questions. The manuscript is well written and well structured. The figures shown are well prepared highlighting the most important results. Overall I recommend publishing this manuscript with some minor revisions.

**Comments**

I find the methods used appropriate for this type of study. However, part of the analysis could be summarized in a Taylor diagram (using the cRMSE). This has the benefit of adding the standard deviation to the evaluation in a format which is easily evaluated graphically. This metric is otherwise not analysed. So I would like to see either adding the standard deviation to the analysis separately or included in a Taylor diagram.

While we agree that Taylor diagrams are an effective way of summarizing multiple metrics on a single plot, the fact we are analyzing the performance of WRF and ERA-5 with height would require a large number of Taylor diagrams, so we prefer to stick with the vertical profiles of each error metric as in Figures 5 and 6.

Instead, we have added the standard deviation as an additional performance metric as suggested. We have updated Figures 5 and 6 (and the corresponding ones in the SI), and the text in many parts of the paper. In summary, WTK-LED outperforms ERA-5 at the offshore site, whereas no clear winner is observed at SGP, with similar considerations holding in all stability conditions.

The authors study the diurnal cycle in more detail and shows that the WRF simulations yields a larger diurnal variability compared to observations, whereas ERA-5's diurnal variation is underestimated. Variations on additional time scales could also be added to the analysis by e.g. computing the Fourier spectrum for the time series for the three different datasets and the two sites. Please consider this in the revision.

The following plot shows the spectrum at the E05 lidar location. The peaks that are statistically significant are the 24-hour one (already analyzed in the paper), and a region corresponding to 4-6 day timeframe, which might be connected to the impact of weather systems. As such, it would be hard to calculate an "average weather-related cycle" from the various data sources to build a new plot parallel to Figures 7 and 10.

[Figure]

**Minor comments**

L42, Molina reference lacks year
We have added the year (2021) to the reference.

L47, Please add a few sentences commenting the Sheridan (2020) results here. Similar studies has also been performed for the North Sea and the Baltic Sea (Kalverla et al 2019, Wind Energy Sci. 2019, 4, 193–209, Hallgren et al. 2020, Energies, 13, 3670; doi:10.3390/en13143670).
We have added the following sentences: "Sheridan et al. (2020) recently validated three reanalysis products using data at one single height from a floating lidar in the U.S. Eastern Seaboard, and found that ERA-5 had the best performance out of the four considered reanalysis products. Similar validations have been performed in Northern Europe, especially focusing on low-level jet events (Kalverla et al., 2019; Hallgren et al., 2020)."

L93, Please elaborate on the sensitivity of assuming w=0 for the horizontal wind sped estimate using this assumption? Will this be significant e.g. during strong convection?
Thanks for catching this. As shown in Werner et al. 2005, assuming $w = 0$ is actually not necessary in this context, so we have removed that sentence. On the other hand, assuming horizontal homogeneity is necessary, but this is expected to mostly impact wind speed retrievals in complex terrain (see for example Bingöl et al. 2008), and not offshore or in flat terrain as considered in our analysis.

- Werner C. (2005) Doppler Wind Lidar. In: Weitkamp C. (eds) Lidar. Springer Series in Optical Sciences, vol 102. Springer, New York, NY. https://doi.org/10.1007/0-387-25101-4_12
- Bingöl, F., Mann, J., & Foussekis, D. (2009). Conically scanning lidar error in complex terrain. *Meteorologische Zeitschrift*, *18*(2), 189-195. https://doi.org/10.1127/0941-2948/2009/0368

L95, Not sure I understand the details here. Is it correct that you get 1 wind speed sample from the lidar every 15 minutes? The hourly estimate is then an average of 4 15 minutes estimates?
Yes, this is correct.

L107, definition of near neutral: with your definition this leaves near neutral to L=0 or L>200 or L<-200. Normally, you would define a range around z/L=0 (see e.g. Sorbjan and Grachev. Boundary-Layer Meteorol 135, 385–405 (2010). https://doi.org/10.1007/s10546-010-9482-3). Please comment and revise

We have now included L=0m in the stable classification (however, that never occurred in the datasets, so the analysis itself did not change). For the rest of the classification, this is consistent with the "range around z/L=0" you pointed out.

L113, What type of lidars where deployed at the offshore location and how was the wind speed evaluated from these? Did you also here get hourly average?

We have added the following information: "The New York State Energy Research and Development Authority (NYSERDA) recently deployed ZephIR ZX300M lidars and made their data publicly available \citep{nyserda2020}." and "The lidars measure at a nominal 50-Hz resolution, and observations are provided as hourly averages, after proprietary quality checks are applied to the data."

L134, Please comment on the different model setup for the land and offshore location

We have added the following comment: "All the main setups that have been shown to have a major impact on modeled wind speed (e.g., the choice of the planetary boundary layer scheme and of the atmospheric forcing) are the same between the offshore and land-based domains. For some other setups, different choices were made between the two domains in order to optimize and tailor the numerical simulations to the specific needs of each domain."

L263, strictly, the conclusions part is more written as "summary and conclusion".

We have changed the title of this section as suggested.

---

## Referee Report (RR1)

I recommend accepting with technical corrections. Although the scope is narrow and the results have limited novelty, the study is valuable for its addition to existing knowledge and for highlighting two sites that are important for the ongoing development of wind energy. I think the authors have sufficiently addressed the issues raised by the referees and editor. Some language and technical issues persist and should be fixed before publishing (see below). The most critical problem is that Fig. 11 was not rendered correctly in the "Author's tracked changes" PDF, which made it impossible to see all the details of the figure correctly.

**1 Specific comments**

- P1-P2 "Dataset" and "data sets" are used, perhaps "datasets" would be more consistent with "dataset" or vise versa?

- P8L157 wind resource → wind resources

- P9L164 use → used

- General language comment: mix of past and present tense

- P12L227: ERA's → ERA-5's

- P17L300: consistently → consistent?

- Fig. 11 is not rendered correctly in the PDF, so I cannot fully review it

- P20L299-L300: Two sentences start with "On the other hand"

- "On the other hand" and "slight/slightly" are used frequently. Using more variation in the language could benefit the text. Just a suggestion.

---

## Referee Report (RR2)

Review of revised version WES 2021-07

I am satisfied with the revisions and responses to my first set of comments.

However, in the latest version where the surface heat fluxes have been introduced I have some further comments.

Comments

Section 2: Since you now also include observational data of sensible and latent heat fluxes you need to present the instrumentation and methods used here as well.

L304: I think you also need to introduce a motivation here to why soil moisture needs to be accurately represented in WRF in order to correctly model the diurnal cycle of the wind speed. Otherwise, this reasoning seems a bit out of context.

L309-310: Not sure I quite follow this. You state that WTK-LED's slightly underpredicts the hub height winds, but previous results all show a positive bias. Please elaborate on what you mean here.

L329 Sentence starting with "On the other hand, the WTK-LED negative bias…". I suggest replacing with "Additionally, the WTK-LED negative bias…"

---

## Author Response (AR2)

*In this document, the editor's comments are in black, the authors' responses are in red.*

We thank the editor for her thoughtful and productive comments.

Dear Dr Pronk and co-authors,

I have read your answers to the reviewer's comments, and I find that you have not entirely addressed their suggestions. The manuscript is partly a repetition of what others have done on many sites for only two locations in the USA.

We would like to explain our choice of the two US locations selected for the analysis, which are indeed limited in number, but they represent sites of primary importance for current and future wind energy development in the US:

1) the offshore location is where the vast majority of the near-future offshore wind energy development will occur;
2) the land-based location is surrounded by many wind plants, a long-term atmospheric observatory, has hosted a field campaign (LABLE) in the last decade, and will host an international field campaign (AWAKEN) starting next year.

We have edited the text to better communicate the reasons behind our choice.

You display the results, but there is little attempt to explain or discuss their implications.

We have created a separate section to present our analysis of the wind plant wake impacts at SGP, which we think contribute to the WRF overestimation of nighttime winds at the site. We think that having a separate section, with slightly stronger text, will help the reader follow our attempt to explain the results presented in the sections before.

Also, we have added comments from the analysis of the surface heat fluxes at SGP, as suggested in a later comment.

The request of comparing the wind direction distributions for the sites to the ones simulated by ERA5 and WRF is reasonable.

We have added the following section to describe how the considered error metrics vary as a function of wind direction:

**3.3 Impact of wind direction**

Different wind direction regimes could also have an impact on the relative performance of WTK-LED and ERA-5. In general, we find that both WTK-LED and ERA-5 are capable of accurately representing the observed wind direction distributions at the considered locations (see histograms in the supplementary materials). At SGP, the wind is mainly from the south or north (wind rose in Figure 2), with a strong seasonal variability as summer months experience mostly southerly flow, whereas in winter a broader range of wind directions is observed. At the offshore location, the winds are mostly from the northwest or the southeast (wind rose in Figure 3), once again with a significant seasonal variability (as described in Bodini et al. (2019)), with summer seeing mostly southwesterly winds, and winter experiencing mostly northwesterlies. Based on these dominant wind regimes, we show in Figure 7 vertical profiles of the five performance metrics with data segregated by wind direction. At SGP,

[Figure]

**Figure 7.** Vertical profiles of performance metrics segregated by dominant wind direction regimes at the SGP C1 site (left) and at the location of the E05 lidar (right). Results from the E06 lidar are included in the Supplementary Materials.

we include northerly (315-45°) and southerly (135-225°) flow, at the E05 lidar we show results for northwesterly (270-360°) and southwesterly (180-270°) winds.

At SGP, both WTK-LED and ERA-5 show a better performance for northerly flow, when looking at bias, cRMSE and EMD. In terms of correlation, southerly flow cases are slightly better for both data sources, but with a very limited difference. As will be detailed in Section 3.5, the worse performance for southerly flow at the site can be connected to potential wind turbine and wind plant wake effects, which impact the SGP observational site for southerly winds.

At the offshore site, WTK-LED shows better skills when modeling southwesterly flow, whereas ERA-5 is more accurate at representing northwesterly flow. This is true across all performance metrics.

Also, in the SI, we have included histograms which compare the distributions of hub-height wind direction from lidar, WRF, and ERA-5.

[Figure]

**Figure S3.** Histogram of 100-m wind direction from lidar data, WRF and ERA-5 at the SGP location.

[Figure]

**Figure S4.** Histogram of 100-m wind direction from lidar data, WRF and ERA-5 at the E05 (top) and E06 (bottom) lidar locations.

Understanding the overprediction of the diurnal cycle over the land location will be very valuable. DTU has done WRF simulations in South Africa (https://orbit.dtu.dk/en/publications/mesoscale-and-microscale-downscaling-for-the-wind-atlas-of-south-) and compared them to the data at 19 62-m masts. We find that the overprediction of the daily cycle occurs in some sites but not others. We have difficulties identifying the physical causes of the errors because of the lack of surface fluxes. In your study, you have such data available at the land site. Perhaps there is a connection to the simulation of the soil moisture and surface fluxes, as suggested by a recent manuscript by Xia et (2021) (DOI: 10.1175/MWR-D-20-0363.1)? I suggest you try further to diagnose the causes of the daily cycle errors and incorporate the results and interpretation in your manuscript.

We have analyzed the sensible and latent heat fluxes from observations and from WRF at the SGP site as suggested, and added the following section to the paper:

**3.6 Impact of surface fluxes at land-based site**

The ability of WTK-LED of accurately representing soil moisture could also contribute explaining the overestimation of the diurnal cycle at SGP. In Figure 12, we compare the average diurnal cycle in sensible and latent heat fluxes from near-surface observations at SGP and from the WTK-LED. We find that WTK-LED overestimates the sensible heat flux during the day, while at night the WTK-LED prediction is, on average, quite accurate. The diurnal cycle of latent heat flux, on the other hand, is quite accurately modeled by WTK-LED throughout the day, on average, with a slight delay compared to the observed one. These results suggest that the exaggerated strong sensible heat flux by WTK-LED during the day (potentially caused by issues in the WTK-LED land use/land cover) is consistent with the WTK-LED's slight underprediction in hub-height wind speed

during the day, as a strong sensible heat flux would increase turbulence and drag. On the other hand, soil moisture seems accurately modeled in WTK-LED given the relatively accurate latent heat flux values, so that the effects on the exaggerated diurnal cycle in hub-height wind speed are likely negligible Xia et al. (2021).

[Figure]

**Figure 12.** Observed and WTK-LED-modeled average diurnal cycle of sensible and latent heat fluxes at the SGP site.

Moreover, as mentioned above, we think that wake effects are related to the strong WRF overestimation of wind speed at night, as confirmed by the directional results added to the paper. These effects are discussed in Section 3.5.
All these considerations have been added to the Conclusions, too.

There are also a few other details to be sorted:
1. Some references are incomplete, e.g. "Vaterite ˘ı: On the stabilization of the general linear group over a ring." No year, no source. Other references are missing their DOI.
   We have carefully reviewed the references and added missing information where needed. We note that we could not find a DOI for the paper by Babíc et al. 2012.

2. The statement "ERA-5 data are publicly available from the ECMWF's MARS archive." is incorrect. The ERA5 model-level data is available from the MARS system; the pressure data is available from the Copernicus website.
   We have fixed this.

---

## Author Response (AR3)

*In this document, the editor's comments are in black, the authors' responses are in red.*

We thank the editor for her thoughtful and productive comments.

I recommend accepting with technical corrections. Although the scope is narrow and the results have limited novelty, the study is valuable for its addition to existing knowledge and for highlighting two sites that are important for the ongoing development of wind energy. I think the authors have sufficiently addressed the issues raised by the referees and editor. Some language and technical issues persist and should be fixed before publishing (see below). The most critical problem is that Fig. 11 was not rendered correctly in the "Author's tracked changes" PDF, which made it impossible to see all the details of the figure correctly.

**Specific comments**

P1-P2 "Dataset" and "data sets" are used, perhaps "datasets" would be more consistent with "dataset" or vise versa? We have fixed this issue.

P8L157 wind resource → wind resources Changed.

P9L164 use → used See our answer below.

General language comment: mix of past and present tense We have revised this. In general, we now use present tense to describe our work, and past tense to describe work done by others in the past.

P12L227: ERA's → ERA-5's Changed.

P17L300: consistently → consistent? Changed to "which is consistent".

Fig. 11 is not rendered correctly in the PDF, so I cannot fully review it Sorry about this, the figure now appears in the latest version of the manuscript.

P20L299-L300: Two sentences start with "On the other hand" We changed one of the two sentences as suggested by the other reviewer.

"On the other hand" and "slight/slightly" are used frequently. Using more variation in the language could benefit the text. Just a suggestion. Thanks for this. We have revised a few sentences. "On the other hand" is now used 5 times throughout the paper. "slight/slightly" is now used 7 times.

*In this document, the editor's comments are in black, the authors' responses are in red.*

We thank the editor for her thoughtful and productive comments.

I am satisfied with the revisions and responses to my first set of comments.

However, in the latest version where the surface heat fluxes have been introduced I have some further comments.

**Comments**

Section 2: Since you now also include observational data of sensible and latent heat fluxes you need to present the instrumentation and methods used here as well.
We have added the following sentence, which includes two references with details on the specific instrument and approach used to retrieve fluxes: "Finally, we consider measurements of sensible and latent heat fluxes from a flux station at the C1 site, which includes a sonic anemometer and an infrared gas analyzer, at 4 m AGL. Details on the calculation of the fluxes from these instruments are described in Fischer (2004), and follow the correction proposed in Webb et al. (1980)."

L304: I think you also need to introduce a motivation here to why soil moisture needs to be accurately represented in WRF in order to correctly model the diurnal cycle of the wind speed. Otherwise, this reasoning seems a bit out of context.
We have added the following sentence: "The ability of WTK-LED of accurately representing soil moisture could also contribute explaining the overestimation of the diurnal cycle at SGP, as the variability of surface fluxes can change atmospheric turbulence and, as a consequence, drag (Geernaert, 1990).".

L309-310: Not sure I quite follow this. You state that WTK-LED's slightly underpredicts the hub height winds, but previous results all show a positive bias. Please elaborate on what you mean here.
We have specified "… is consistent with the WTK-LED's slight underprediction in the mean diurnal cycle of hub-height wind speed between 16-23 UTC"

L329 Sentence starting with "On the other hand, the WTK-LED negative bias...". I suggest replacing with "Additionally, the WTK-LED negative bias..."
Changed.